# Dopamine modulates subcortical responses to surprising sounds

**Catalina Valdés-Baizabal**[1,2☯], **Guillermo V. Carbajal**[1,2☯], **David Pérez-González**[1,2]*, **Manuel S. Malmierca**[1,2,3]*

1 Cognitive and Auditory Neuroscience Laboratory (CANELAB), Institute of Neuroscience of Castilla y León (INCYL), Salamanca, Spain, 2 Institute for Biomedical Research of Salamanca (IBSAL), Salamanca, Spain, 3 Department of Biology and Pathology, Faculty of Medicine, University of Salamanca, Salamanca, Spain

☯ These authors contributed equally to this work.
* davidpg@usal.es (DPG); msm@usal.es (MSM)

**Data Availability Statement:** All relevant data are within the manuscript and its Supporting Information files.

**Funding:** Financial support provided by Spanish MINECO (SAF2016-75803-P) to MSM. CVB held a

## Abstract

Dopamine guides behavior and learning through pleasure, according to classic understanding. Dopaminergic neurons are traditionally thought to signal positive or negative prediction errors (PEs) when reward expectations are, respectively, exceeded or not matched. These signed PEs are quite different from the unsigned PEs, which report surprise during sensory processing. But mounting theoretical accounts from the predictive processing framework postulate that dopamine, as a neuromodulator, could potentially regulate the postsynaptic gain of sensory neurons, thereby scaling unsigned PEs according to their expected precision or confidence. Despite ample modeling work, the physiological effects of dopamine on the processing of surprising sensory information are yet to be addressed experimentally. In this study, we tested how dopamine modulates midbrain processing of unexpected tones. We recorded extracellular responses from the rat inferior colliculus to oddball and cascade sequences, before, during, and after the microiontophoretic application of dopamine or eticlopride (a $D_2$-like receptor antagonist). Results demonstrate that dopamine reduces the net neuronal responsiveness exclusively to unexpected sensory input without significantly altering the processing of expected input. We conclude that dopaminergic projections from the thalamic subparafascicular nucleus to the inferior colliculus could encode the expected precision of unsigned PEs, attenuating via $D_2$-like receptors the postsynaptic gain of sensory inputs forwarded by the auditory midbrain neurons. This direct dopaminergic modulation of sensory PE signaling has profound implications for both the predictive coding framework and the understanding of dopamine function.

## Introduction

Dopamine is commonly regarded as the modulatory neurotransmitter underlying phenomenological experiences such as pleasure and joy. This "hedonic" impression derives from classic empirical approaches emphasizing the role of dopamine in the anticipation and seeking of rewarding outcomes [1–3]. Indeed, mounting evidence supports that dopamine regulates

grant from Mexican CONACYT (216652). GVC held a fellowship from the Spanish MICINN (BES-2017-080030). The funders had no role in study design, data collection and analysis, decision to publish, or preparation of the manuscript.

**Competing interests:** I have read the journal's policy and the authors of this manuscript have the following competing interests: MSM is an Academic Editor and a member of editorial board of PLOS Medicine.

**Abbreviations:** Aq, aqueduct; CAS, cascade sequence; CoIC, commissure of the inferior colliculus; CSI, common stimulus-specific adaptation index; DEV, deviant condition; FR, firing rate (spikes/s); FRA, frequency response area; HCN, hyperpolarization-activated cyclic nucleotide-gated; IC, inferior colliculus; iMM, index of neuronal mismatch; LDT, laterodorsal tegmental; NMDA, N-methyl-D-aspartate; PAG, periaqueductal gray; PE, prediction error; PPT, pedunculopontine tegmental; SFR, spontaneous firing rate (spikes/s); SPF, subparafascicular nucleus of the thalamus; SSA, stimulus-specific adaptation; STD, standard condition.

movement, motivation, and learning by tracking the violations of our reward expectations, which are encoded as reward prediction errors (PEs) [4–6]. Whereas sensory PEs report the surprise of unexpected sensory inputs in absolute magnitudes, reward PEs indicate whether outcomes were better or worse than expected, resulting in positively and negatively signed PEs, respectively [7]. As a rule of thumb, dopaminergic neurons report positive PE values by increasing their phasic firing of action potentials and negative PE values by reducing their tonic discharge rates [4–6]. Hence, these dopaminergic neurons seem to signal a motivational ambivalence that guides adaptative motor, learning, and decision-making processes.

But some experimental evidence defies this classic understanding of dopamine. Aversive outcomes and cues that predict them can elicit dopaminergic activity [8–12], as well as events in which the reward PE should theoretically be zero, such as unexpected or surprising stimuli [12–18]. These dopaminergic responses seem to report sensory stimuli that may have behavioral relevance and should trigger an appropriate coordinated response, thus encoding perceptual salience without any positive or negative value.

The existence of signed and unsigned PEs is not mutually exclusive, and nowadays, there is relative consensus that dopamine participates in the attribution of physical and surprise salience to sensory stimuli (for a recent and comprehensive review, please refer to [19]). Some authors suggest that novel and physically salient stimuli might be inherently rewarding, as they provide new information, which could be of value for adaptive behavior [20,21]. Other works postulate a dual dopaminergic signaling, which respectively reports surprise and hedonic value in parallel [22,23]. But proposals from the predictive processing framework advocate for a more integrative account of dopaminergic function, not necessarily bound to reward processing [24,25].

The predictive coding framework comprises biologically informed Bayesian models based on early cybernetic theories, which regard the brain as a predictive Helmholtz machine [26–28]. Because we cannot directly access the "true" external world, but only its impression in our sensorium, the brain must infer the cause of those sensations. The brain generates expectations about the "hidden" states of the world by means of a hierarchical generative model, in which higher neural populations try to explain away (i.e., predict and inhibit) input from lower neural populations, and the resulting PE is used to update prior beliefs (i.e., learning). At each processing level, excitatory neurons receive excitatory input conveying bottom-up sensory evidence, as well as inhibitory input conveying top-down expectations. When these 2 inputs are congruent, their postsynaptic potentials cancel out; but when they are incongruent, a PE is generated to report the mismatch [29–33].

This unsigned PE accounts for both perception and learning, so the predictive processing framework can smoothly accommodate dopaminergic responses elicited by surprise. Behavior is optimal, not when reward (positive PE) is maximized, but when surprise (unsigned PE) is minimized, as this keeps the organism from potentially harmful interactions with the environment. Minimal PE is pursued via perceptual inference, i.e., improving the internal model of the world to better explain away incoming sensory input, and via active inference, i.e., changing sensory input by engaging in actions with predictable outcomes [27,28]. Classic reinforcement learning tasks may have confused dopaminergic responses with reward PEs because of the more generic role that dopamine plays in PE minimization. Cues predicting rewards minimize PE by resolving the uncertainty about future outcomes, which is flagged by dopamine release [24,25].

From the predictive processing standpoint, there are only 2 sorts of things that need to be inferred about the world: the state of the world and the uncertainty about that state [24]. Beliefs about the hidden states of the world emerge from the hierarchical exchange of top-down predictions and bottom-up PEs, embodied in the synaptic activity of the nervous system. But

every inferential process entails a certain degree of uncertainty due to, e.g., our sensory limitations, ambiguity, noise, or volatility in the probabilistic structure of the environment. Such uncertainty is accounted for in terms of posterior confidence or expected precision by means of the postsynaptic gain. Hence, synaptic messages are weighted according to their expected precision as they are passed along the processing hierarchy. When expected precision is high, PE signals are deemed reliable and receive postsynaptic amplification to strengthen their updating power. Conversely, when low precision is expected, PE signals undergo attenuation to prevent misrepresentations, and the influence of prior beliefs becomes prominent. In plain words, the expected precision is the postsynaptic gain that scales PE to modulate its influence on higher processing levels, such that more is learned from precise PEs than from noisy and unreliable PEs [33].

Neuromodulators, such as dopamine, cannot directly excite or inhibit postsynaptic responses but only weight the postsynaptic responses to other neurotransmitters, acting as a gain control mechanism. Therefore, the only possible function of the dopaminergic system is to encode the expected precision [24,25], playing a role in both perceptual and active inference by conferring contextual flexibility to both sensory and motor processing. Dopamine influence in motor processing and its function in active inference have received considerable scientific attention [24,34–37]. The same cannot be said about perceptual inference, maybe because of the relatively sparse and diffuse dopaminergic innervation of sensory structures. As a consequence of scarce empirical research, the physiological effects of dopamine on the processing of surprising sensory information are not well understood yet [38].

The processing of surprising sensory information has been classically studied in humans using the auditory oddball paradigm [39], in which the successive repetition of a tone ("standard condition" [STD]) is randomly interrupted by a surprising oddball tone ("deviant condition" [DEV]). When applied to animal models, the oddball paradigm unveils a phenomenon of neuronal short-term plasticity called stimulus-specific adaptation (SSA), measured as the difference between DEV and STD responses [40]. SSA has been traditionally regarded as a rather mechanistic product of synaptic fatigue specifically affecting the transmission of the STD signal along the auditory system, while the processing channels conveying the infrequent DEV signal remained fresh [41,42]. Nevertheless, the predictive processing framework has also reinterpreted the SSA observed in the auditory system [43,44]. As higher neural populations explain away bottom-up sensory information, lower neural populations receiving their top-down predictions decrease responsiveness to expected sensory inputs, which, during an oddball paradigm, manifests functionally as SSA of the STD response. But when encountering DEV stimuli, predictions fail, forwarding PE signals to report the unexpected sensory input to the higher processing levels and update prior beliefs, thus generating a larger DEV response.

In a previous SSA study from our lab performed in awake and anaesthetized rodents, we demonstrated that DEV responses of auditory neurons from the midbrain, thalamus, and cortex were better explained as PE signaling activity [45]. SSA first emerges in the auditory system at the level of the inferior colliculus (IC), mainly in its nonlemniscal portion (i.e., the IC cortices) [46]. As a site of convergence of both ascending and descending auditory pathways, the IC plays a key role in processing surprising sounds [47] and shaping the auditory context [48]. The complex computational network of the IC integrates excitatory, inhibitory, and rich neuromodulatory input [49,50], which includes dopaminergic innervation from the thalamic subparafascicular nucleus (SPF) [51–58]. Previous reports have detected mRNA coding for dopaminergic $D_2$-like receptors in the IC [57,58] and proved its functional expression as protein. Other studies have confirmed that dopamine modulates the auditory responses of IC neurons in heterogeneous manners [52,59]. However, the involvement of dopaminergic modulation of SSA and sensory PE signaling in the IC is yet to be proven.

In order to test whether dopamine can modulate surprise responses and predictive sensory processing, we performed microiontophoretic injections of dopamine and eticlopride (a $D_2$-like receptor antagonist) in the nonlemniscal IC while recording single and multiunit responses under oddball and regular auditory sequences. Our results demonstrate that dopamine has a profound effect on how unexpected sounds are processed, presumably encoding expected precision of sensory PEs at the level of the auditory midbrain.

## Results

In order to study the role of dopamine in shaping SSA and sensory PE signaling in the nonlemniscal IC, we recorded the auditory responses from a total of 142 single and multiunits in 31 young adult Long–Evans rats. In a first series of experiments, we presented the oddball paradigm before, during, and after microiontophoretic application of dopamine ($n$ = 94) or eticlopride ($n$ = 43). DEV and STD responses were used to calculate a common SSA index (CSI) for each unit and condition (see Protocol 1 in Materials and methods). In an additional series of experiments ($n$ = 43, from the former pool of units), we also presented 2 predictable cascade sequences (CASs) in addition to the oddball paradigm, following the methodology of a previous study [45]. In this subset, we measured SSA using the index of neuronal mismatch (iMM), which is calculated including DEV, STD, and CAS responses (see Protocol 2 in Materials and methods). Notwithstanding, both the iMM and the CSI are largely equivalent for indexing SSA [45].

Histological verification located all recording sites in the rostral cortex of the IC (Fig 1). This subdivision is part of the nonlemniscal IC, where SSA indices tend to be higher [46,47,60].

### Dopamine effects on the CSI

The microiontophoretic application of dopamine-induced changes in the firing rate (FR) and frequency response area (FRA) of the recorded units (Fig 2). As a general consequence, SSA indices decreased by 15% in our sample, falling from a median CSI of 0.51 (0.25–0.78) in the control condition to 0.43 (0.15–0.73) after dopamine application ($p$ = 0.014; Fig 3A). Such SSA reduction was caused by a 26% drop in the median DEV response (control FR: 5.63 [3.25–10.00] spikes/s; dopamine FR: 4.19 [1.63–7.63]; $p$ < 0.001; red in Fig 3B and 3C), whereas the STD response did not show a significant change (control FR: 1.71 [0.64–3.83]; dopamine FR: 1.63 [0.36–3.65]; $p$ = 0.284; blue in Fig 3B and 3D). As observed in previous reports [46], the spontaneous firing rate (SFR) found in the nonlemniscal IC tended to be very scarce (for individual examples, see Fig 3E and 3F) and did not change significantly with dopamine application (control SFR: 0.18 [0.05–0.77]; dopamine SFR: 0.15 [0.03–0.90]; $p$ = 0.525; gray in Fig 3B).

Previous studies had reported heterogeneous dopaminergic effects on the response of IC neurons [52,59], so we performed a bootstrap analysis to evaluate the statistical significance of CSI changes unit by unit. We confirmed such heterogeneity across our sample, with 42 units following the population trend by decreasing their CSI, whereas 33 units showed CSI increments under dopamine; 19 units remained unaltered (Fig 3A). Fig 3E shows the response of a unit to STD (blue) and DEV (red) in the control condition (left panel), during dopamine application (middle panel), and after recovery (right panel). The application of dopamine caused an increment of the STD response and a decrement of the DEV response, leading to a decrease of the CSI. In contrast, the unit in Fig 3F showed a decrement of the response to both STD and DEV during the application of dopamine, thus resulting in an increase of the CSI. The effects of dopamine peaked around 8–10 minutes after microiontophoretic application, followed by a progressive recovery to baseline values that could take beyond 90 minutes (Fig 3E and 3F, right panels).

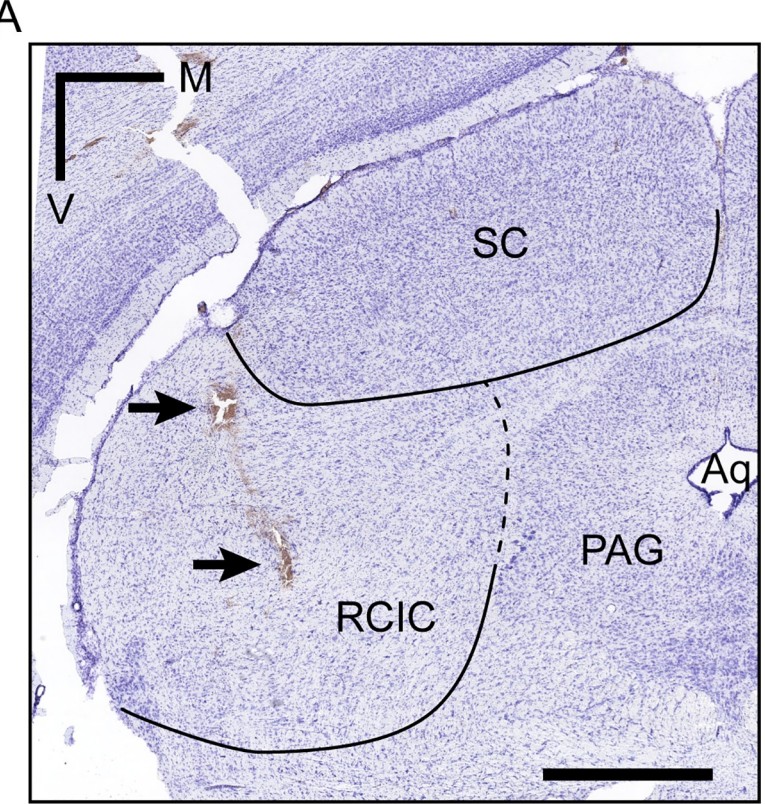

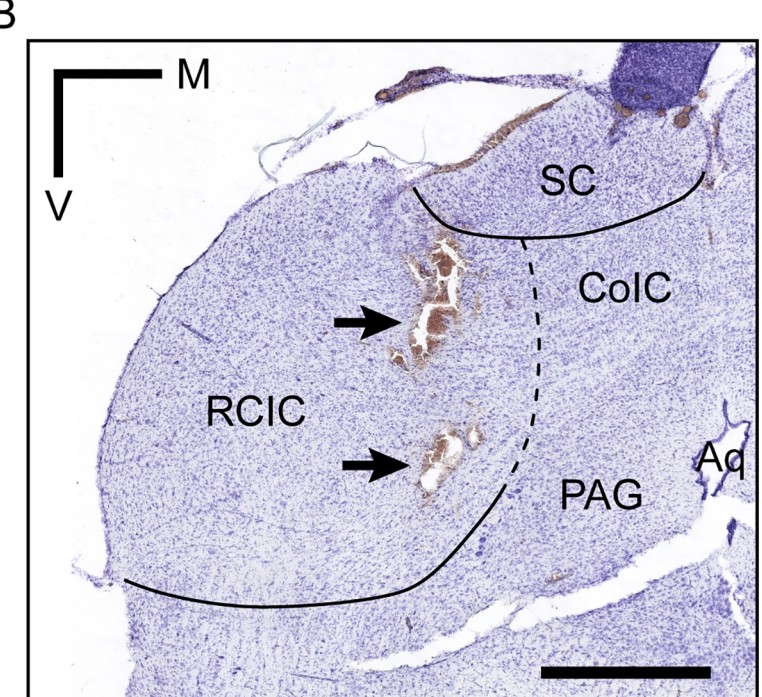

**Fig 1. Histological location. (A, B)** Coronal sections from 2 different animals showing electrolytic lesions (arrows) in the RCIC. Scale bar = 1 mm. Aq, aqueduct; CoIC, commissure of the inferior colliculus; M, medial; PAG, periaqueductal gray; RCIC, rostral cortex of the inferior colliculus; SC, superior colliculus; V, ventral.

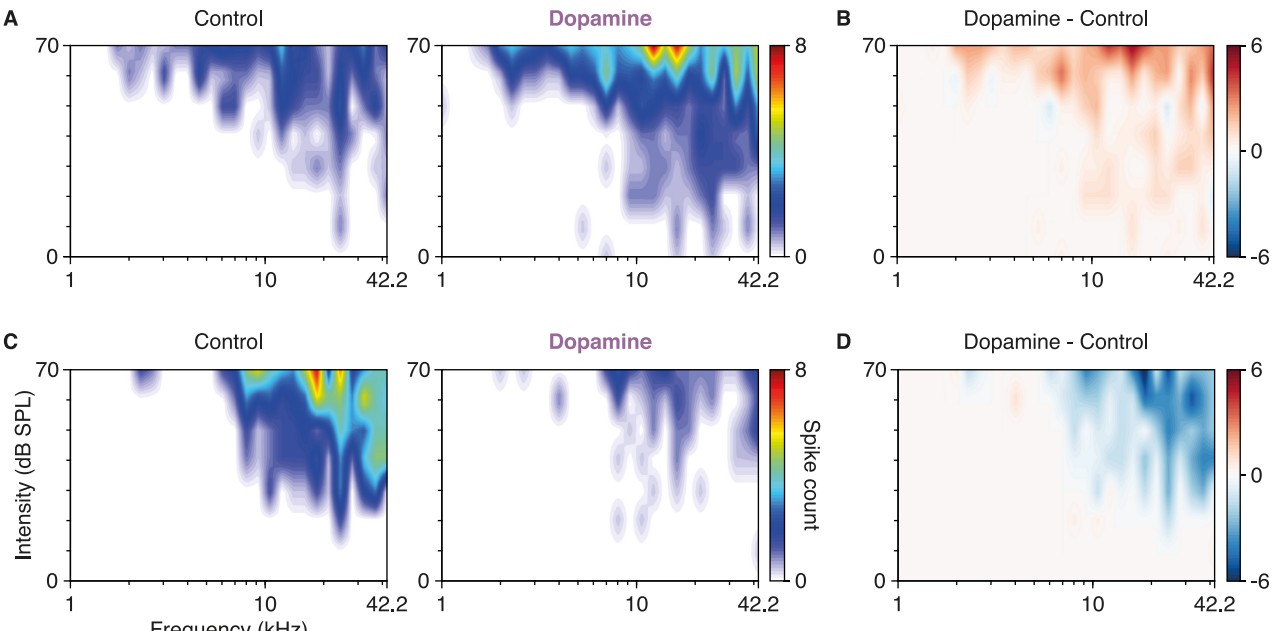

**Fig 2. Dopamine effects on the FRA. (A)** FRA of a neuron in control condition (left panel) and after dopamine application (right panel). **(B)** The subtraction of the control FRA from the FRA after dopamine application reveals that dopamine increased the excitability of this unit. **(C)** FRA of another neuron in control condition (left panel) and after dopamine application (right panel). **(D)** The subtraction of the control FRA from that after dopamine application in **(C)** reveals that dopamine decreased the excitability of this neuron. The underlying data for this figure can be found in S1 Data. dB SPL, decibels of sound pressure level; FRA, frequency response area

## Eticlopride effects on the CSI

We aimed to determine whether dopaminergic effects on the CSI were mediated by $D_2$-like receptors, as suggested by previous reports [57,58]. To test endogenous dopaminergic modulation on SSA mediated by $D_2$-like receptors, we applied eticlopride, a $D_2$-like receptor antagonist, to 43 units. We observed no significant response changes at sample level (DEV FR: $p = 0.609$; STD FR: $p = 0.769$; SFR: $p = 0.405$; CSI change: $p = 0.170$; Fig 4). However, we performed a bootstrap analysis to evaluate the statistical significance of CSI changes in each unit under eticlopride influence, which revealed that only 11 units remained unaffected (Fig 4A). The CSI had significantly decreased in 13 units (see individual example in Fig 4E) and increased in 19 units (see individual example in Fig 4F), implying that eticlopride was indeed antagonizing endogenous dopaminergic modulation mediated by $D_2$-like receptors on those units.

## Dopamine effects on unexpected auditory input

To test whether dopamine modulates sensory PE signaling in the nonlemniscal IC, we performed an additional set of experiments (see Protocol 2 in Materials and methods) adapting our methodology to that of a previous study [45]. Alongside the oddball paradigm (Fig 5A), we recorded responses of 43 units to 2 CAS conditions, which consisted of a sequence of 10 tones presented in a predictable succession of increasing or decreasing frequencies (Fig 5B).

We used a bootstrap analysis to evaluate the statistical significance of the effect of dopamine on the iMM of each recording, which confirmed that 23 units underwent heterogeneous iMM changes (Fig 5C, colored dots, each representing one tested frequency), whereas another 18 remained stable (Fig 5C, gray dots). Results agreed with those obtained using the CSI, as the

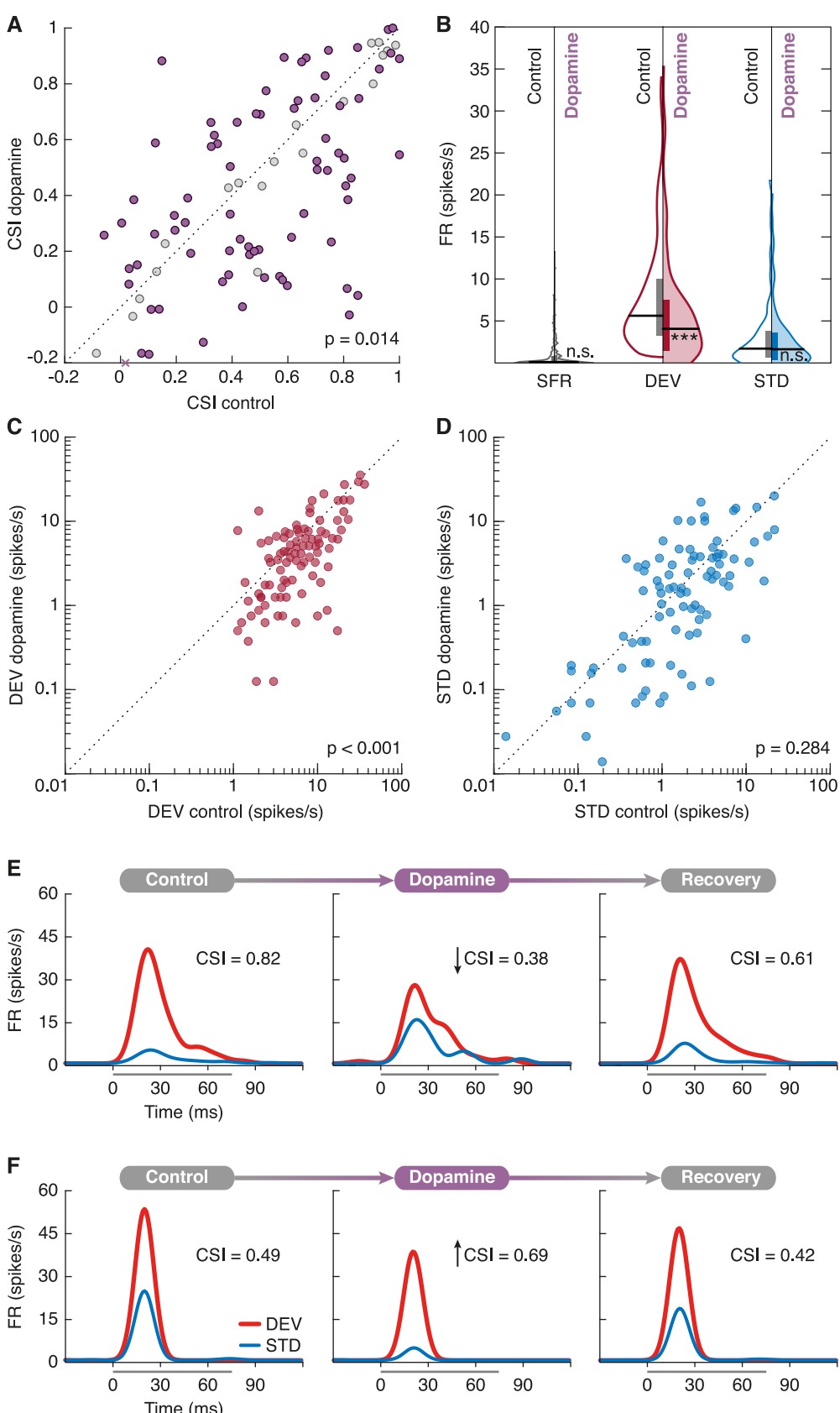

**Fig 3. Dopamine effects on the CSI. (A)** Scatterplot of the CSI in control condition versus dopamine application. Units that underwent significant CSI changes are represented in purple, whereas the rest are marked as gray dots. The purple cross on the abscissa axis represents 1 CSI measurement in which ordinate value falls out of scale ($y = -0.59$). **(B)** Violin plots of the SFR (gray), DEV response (red), and STD response (blue). Control conditions are represented in the left half of each violin (no color), whereas dopamine effects are on display in the right half (colored). Horizontal thick black lines mark the median of each distribution, and vertical bars cover the interquartile range. Regarding statistical significance, n.s. indicates that $p > 0.05$ and *** indicates that $p < 0.001$. **(C)** Scatterplot of DEV responses in control condition versus dopamine application. **(D)** Scatterplot of STD responses in control condition versus dopamine application. **(E)** Peristimulus histogram of a unit before (left panel), during (middle panel), and after (right panel) dopamine application. In this case, dopamine reduced the CSI. **(F)** Another example showing the opposite effects. The underlying data for this Figure can be found in S2 Data. CSI, common stimulus-specific adaptation index; DEV, deviant condition; FR, firing rate; SFR, spontaneous firing rate; STD, standard condition.

iMM of the sample fell by 22%, from a median of 0.57 (0.41–0.69) in the control condition to a median of 0.45 (0.27–0.65) under dopaminergic influence ($p = 0.002$; Fig 5C). This was caused by a significant reduction in the median DEV response (control normalized FR: 0.70 [0.58–0.80]; dopamine normalized FR: 0.64 [0.47–0.78]; $p = 0.002$), whereas the STD response was not affected (control normalized FR: 0.11 [0.02–0.23]; dopamine normalized FR: 0.10 [0.03–0.31]; $p = 0.188$). Most interestingly, CAS responses also remained unaffected by dopamine application (control normalized FR: 0.68 [0.52–0.77]; dopamine normalized FR: 0.71 [0.57–0.83]; $p = 0.115$; Fig 5D).

## Discussion

We recorded single and multiunit activity in the nonlemniscal IC (Fig 1) under an auditory oddball paradigm while performing microiontophoretic applications of dopamine and eticlopride ($D_2$-like receptor antagonist). Following the discovery of PE signaling activity in the nonlemniscal IC [45], we included CASs [61] in a subset of experiments to address dopamine role from a predictive processing standpoint. This resulted in 3 stimulation conditions: (1) STD or expected repetition (Fig 5A, bottom), susceptible of generating intense SSA; (2) DEV or unexpected change (Fig 5A, top), which should be the most surprising and thus elicit the strongest PE signaling; and (3) CAS or expected change (Fig 5B), a condition featuring the same STD-to-DEV step, but which neither undergo SSA (unlike STD) nor should entail a PE (or, at least, not as strong as DEV). Our results revealed that dopamine modulates surprise processing in the auditory midbrain.

### Dopamine attenuates PE signaling from the auditory midbrain

Dopamine application caused a 15% reduction of SSA indices in the nonlemniscal IC (Figs 3A and 5C) because of a general drop in DEV responses of about 25% (Fig 3B and 3C). Neither STD (Fig 3B and 3D) nor CAS responses were significantly affected at population level (Fig 5D). The differential effect of dopamine on DEV and STD cannot be explained by the differences in their control FRs, because CAS yielded FRs as high as DEV that were not similarly reduced by dopamine (Fig 5D). In other words, dopamine application decreased the responsiveness to surprising stimuli, whereas the responsiveness to the expected stimuli remained stable. Therefore, dopaminergic action on the auditory neurons of the nonlemniscal IC exclusively modulates PE signaling.

Eticlopride effects did not describe significant tendencies at population level (Fig 4). Nevertheless, about 75% of our sample manifested significant SSA changes under eticlopride (Fig 4A, colored dots). This confirms the release of endogenous dopamine, as well as the functional expression of $D_2$-like receptors in the nonlemniscal IC. Taken together with previous findings

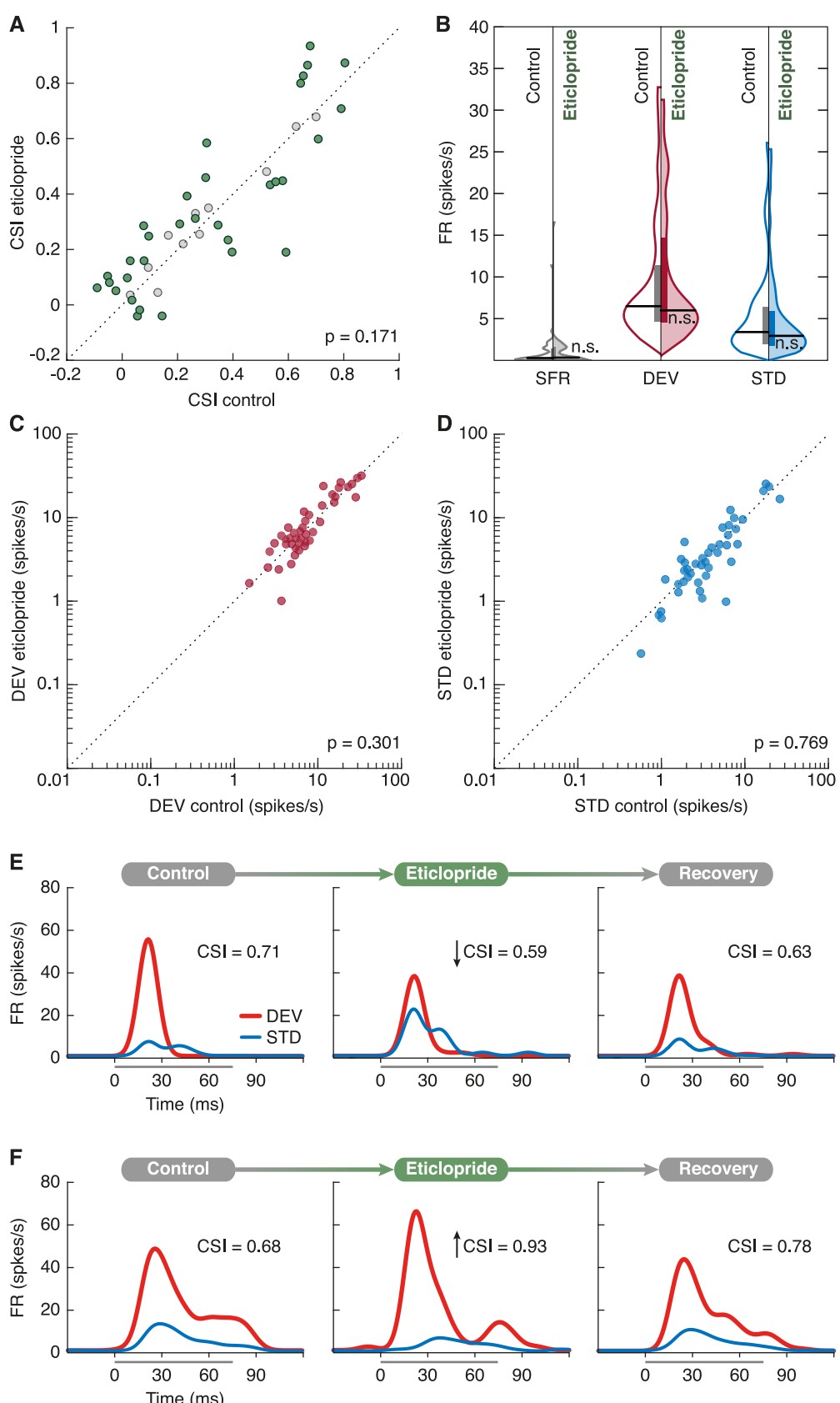

**Fig 4. Eticlopride effects on the CSI. (A)** Scatterplot of the CSI in control condition versus eticlopride application. Units that underwent significant CSI changes are represented in green, whereas the rest are marked as gray dots. **B.** Violin plots of the SFR (gray), DEV response (red), and STD response (blue). Control conditions are represented in the left half of each violin (no color), whereas eticlopride effects are on display in the right half (colored). Horizontal thick black lines mark the median of each distribution, and vertical bars cover the interquartile range. Regarding statistical significance, n.s. indicates that $p > 0.05$. **(C)** Scatterplot of DEV responses in control condition versus eticlopride application. **(D)** Scatterplot of STD responses in control condition versus eticlopride application. **(E)** Peristimulus histogram of a unit before (left panel), during (middle panel), and after (right panel) eticlopride application. In this case, eticlopride reduced the CSI. **(F)** Another unit example showing the opposite effects. The underlying data for this figure can be found in S3 Data. CSI, common stimulus-specific adaptation index; DEV, deviant condition; FR, firing rate; SFR, spontaneous firing rate; STD, standard condition.

regarding dopaminergic modulation of the IC [52,57–59], the attenuation of PE signaling is most likely mediated by $D_2$-like receptors.

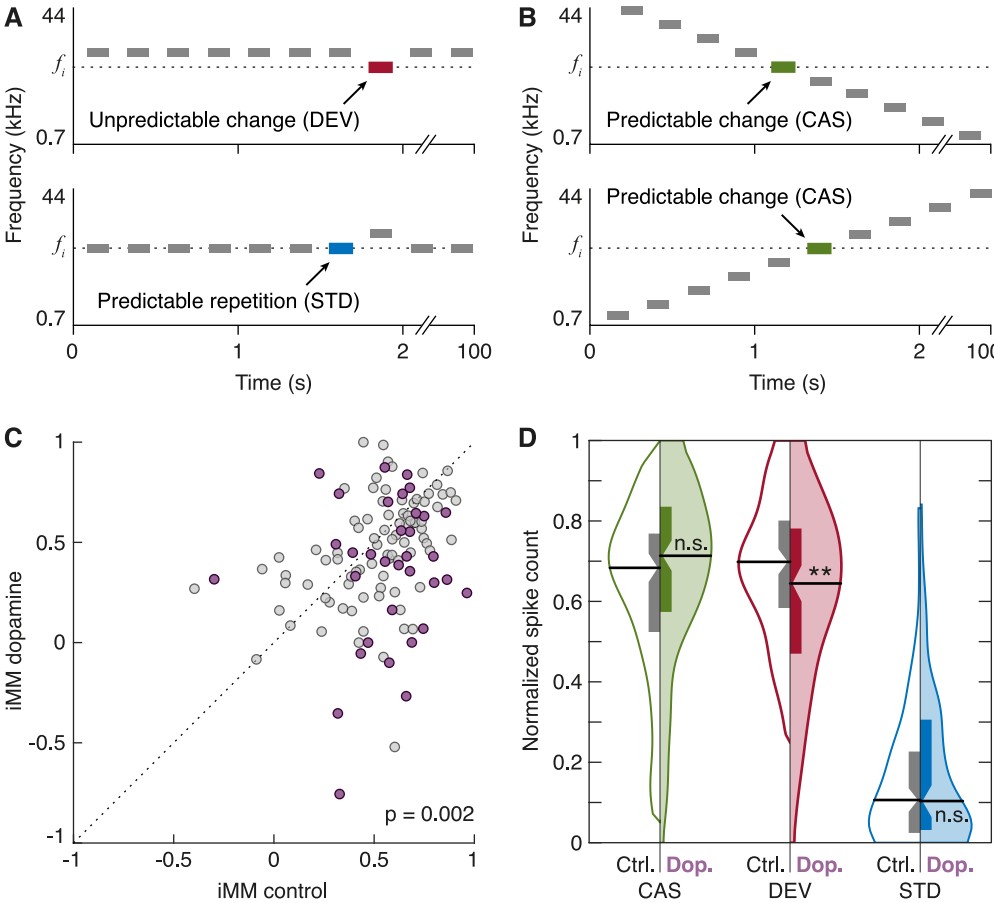

**Fig 5. Dopamine effects on unexpected auditory input. (A)** Oddball paradigm, displaying 2 experimental conditions for a given $f_i$ target tone. **(B)** CASs highlighting the $f_i$ target tone. **(C)** Scatterplot of the iMM in control condition versus dopamine application. Frequencies that underwent significant iMM changes are represented in purple, whereas the rest are marked as gray dots. **(D)** Violin plots of the CAS (green), DEV (red), and STD (blue) normalized responses. Control conditions are represented in the left half of each violin (no color), and dopamine effects are on display in the right half (colored). Horizontal thick black lines mark the median of each distribution, and the boxplots inside each distribution indicate the interquartile range, with the confidence interval for the median indicated by the notches. Regarding statistical significance, n.s. indicates that $p > 0.05$, and $^{**}$ indicates that $p < 0.01$ (repeated-measures ANOVA, Dunn–Šidák correction). The underlying data for this figure can be found in S4 Data. CAS, cascade sequence; DEV, deviant condition; FR, firing rate; iMM, index of neuronal mismatch; SFR, spontaneous firing rate; STD, standard condition.

The net attenuation of PE signaling from the auditory midbrain under dopamine influence is unique as compared with the effects of other neurotransmitters and neuromodulators on IC neurons. GABAergic and glutamatergic manipulations alter the general excitability of IC neurons, thereby exerting symmetrical effects on both STD and DEV responses [62–64]. Cholinergic and cannabinoid manipulation yield asymmetrical effects that mostly affect STD responses [65,66]. Activation of $M_1$ muscarinic receptors tend to reduce average SSA indices by increasing responsiveness to repetitive stimuli (i.e., STD) [65]. Dopamine delivers asymmetrical effects that also tend to reduce average SSA indices. But the activation of $D_2$-like receptors decreases the responsiveness to surprising stimuli (i.e., DEV). The complementary effects of dopaminergic and cholinergic modulation hint at a conjoint action of neuromodulatory systems in adjusting the bottom-up flow of sensory information form subcortical structures (Fig 6).

## Intrinsic and synaptic properties generate heterogeneous dopaminergic effects

In line with previous reports [52,59], we observed heterogeneous dopaminergic effects across units (see individual examples in Figs 1, 3E, 3F, 4E and 4F). Complex dopaminergic interactions altering the excitation–inhibition balance cannot be accurately tracked, because the exact location and neuronal types expressing $D_2$-like receptors in the IC are yet to be determined. Notwithstanding, the heterogeneity of dopaminergic effects must result from distinctive intrinsic and synaptic properties.

$D_2$-like receptors are coupled to G proteins, which regulate the activity of manifold voltage-gated ion channels, adjusting excitability depending on the repertoire expressed in each neuron [67]. $D_2$-like receptors coupled to $G_{i/o}$ proteins can both increase potassium currents and decrease calcium currents via $G_{\beta\gamma}$ subunit complex, thereby reducing excitability [67]. The opening probability of calcium channels can also diminish by the activation of $D_2$-like receptors coupled to $G_q$ proteins [67]. $D_2$-like receptor activation can augment or reduce sodium currents depending on the receptor subtypes expressed on the neuronal membrane [67]. Furthermore, $D_2$-like receptor activation can also reduce N-methyl-D-aspartate (NMDA) synaptic transmission, decreasing the FR [67]. In addition, nonlemniscal IC neurons express hyperpolarization-activated cyclic nucleotide-gated (HCN) channels [68,69], which can be modulated by dopamine and yield mixed effects on neuronal excitability [70].

On the other hand, dopamine and eticlopride can interact with $D_2$-like receptors expressed in a presynaptic neuron. Both glutamatergic and GABAergic projections converge onto single IC neurons [49,71,72], which may also receive dopaminergic inputs from the SPF [51–55,57]. Dopamine could potentially activate presynaptic $D_2$-like receptors expressed in a excitatory neuron, as described in striatal medium spiny neurons [73–75], or conversely in a inhibitory neuron, as demonstrated in the ventral tegmental area [76].

## SPF dopaminergic projections to the IC could scale sensory PEs

Dopaminergic function has been traditionally studied in the context of reinforcement learning, in which dopamine is thought to report the discrepancies between expected and observed rewards in "reward PEs" [2,3]. Dopaminergic neurons report positive PE values when reward expectations are exceeded by increasing their firing and negative PE values when reward expectations are not matched by reducing their tonic discharge rates, thereby guiding the learning process [5,6]. Hence, these signed "reward PEs" encoded by dopaminergic neurons are substantially different from the unsigned "sensory PEs" encoded by auditory neurons in the nonlemniscal IC [31–33]. According to this classic interpretation of dopaminergic

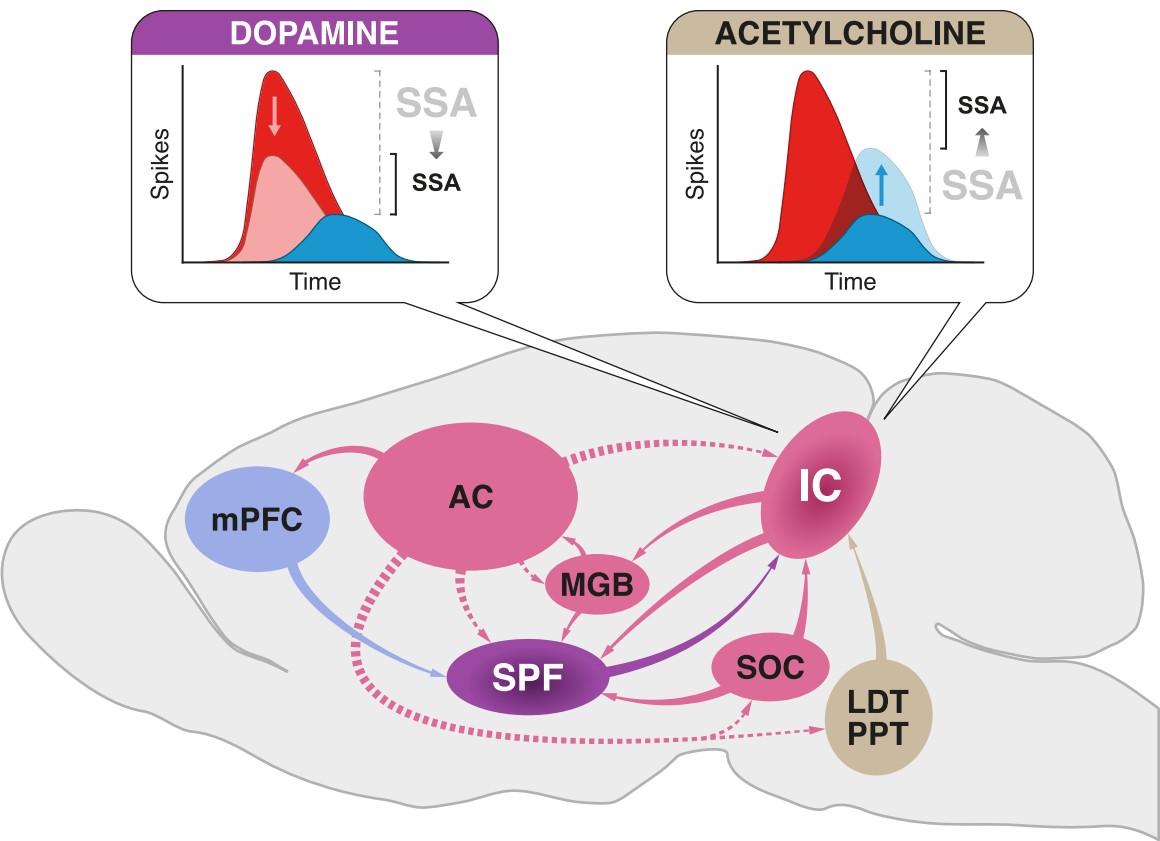

**Fig 6. SPF auditory afferences.** Schematic diagram of the main connections involved in SSA modulation and PE precision-weighting in the IC. The auditory pathway (in pink) is composed of ascending (solid arrows) and descending connections (dashed arrows) between cortical and subcortical auditory structures. The hierarchical exchange of bottom-up PEs and top-down predictions postulated by the predictive coding framework could be extended to subcortical auditory neurons by means of these feedback loops. The SPF (purple) receives input from main auditory nuclei (pink) and from the mPFC (violet), integrating information from manifold hierarchical processing levels. Such connectivity could allow SPF dopaminergic neurons to estimate the volatility of the probabilistic structure of the auditory context. In turn, the SPF sends dopaminergic projections back to the IC to signal the expected precision of PE signaling at low processing levels. Dopamine release in the nonlemnical IC reduces the postsynaptic responses to surprising stimuli (red) but has no effect on repetitive ones (blue), consequently reducing SSA indices. By contrast, cholinergic modulation from the LDT and PPT nuclei (brown) of the brainstem increases the responses to repetitive stimuli in the IC [65]. Note that the net effect of both dopamine and acetylcholine is the reduction of SSA indices, decreasing the relative saliency of surprising input through complementary means. AC, auditory cortex; CAS, cascade sequence; IC, inferior colliculus; LDT, laterodorsal tegmental; MGB, medial geniculate body; mPFC, medial prefrontal cortex; PE, prediction error; PPT, pedunculopontine tegmental; SOC, superior olivary complex; SPF, subparafascicular nucleus of the thalamus; SSA, stimulus-specific adaptation.

function, exogenous dopamine ejections in the nonlemniscal IC should mimic reinforcing signals coming from the SPF [51–55,57]. Speculatively, such dopaminergic input might aim to induce long-term potentiation on IC neurons to build lasting associations between acoustic cues and rewarding outcomes, thereby contributing to establish reward expectations. However, we fail to see why these dopaminergic PEs would mitigate the transmission of sensory PEs from the nonlemniscal IC, as evidenced by the reduced surprise responsiveness we have observed after dopamine application.

The recent interpretation from the predictive processing framework argues that information about the hidden states of the world (i.e., PEs) cannot be encoded by dopamine release, because dopamine cannot directly excite the postsynaptic responses, which would be needed to convey that information [24]. Dopamine can only modulate the postsynaptic responses to other neurotransmitters, a function more compatible with expected precision encoding and

PE weighting. This alternative approach spares the need of 2 distinct types of PE signaling while better accommodating some findings that were not easily explained as reward PEs [24]. A significant portion of dopaminergic neurons increase their firing in response to aversive stimuli and cues that predict them, contrary to how reward PEs should work [8–12]. Most relevant to the present study, some dopaminergic neurons also respond to conditions in which the reward PE should theoretically be zero, including unexpected or surprising stimuli [12–18]. Therefore, we consider that the most suited way of interpreting our data is through the lens of a precision-weighting mechanism. Dopamine function in the auditory midbrain could be to account for the uncertainty about the probabilistic structure of the auditory context and scale sensory PEs accordingly.

The predictive processing framework describes a processing hierarchy in which PEs emitted by a lower-level system become the input for a higher-level system, whereas feedback from the higher-level system provides the expectations for the lower-level system [29–33]. Hence, not all the sensory input is conveyed bottom-up through the processing hierarchy but only the resultant PEs at each level, which, in turn, are progressively explained away level after level. Consequently, expectations in lower-level neural populations are rather overfitted, short-spanned, and prone to PE, whereas higher neural populations acquire greater integrative scope at the expense of accuracy. Thus, higher-order predictions are less specific but also provide more abstract and durable beliefs about the probabilistic structure of the environment and its volatility, which can be used to optimally adjust sensory processing at lower levels.

Nonlemniscal neurons on the IC cortices receive abundant ascending, descending, and neuromodulatory input through their large dendritic arbors [50,55,77,78], which translate into wide auditory receptive fields (Fig 2) capable of integrating frequency fluctuations in the time scale of seconds [79–81]. During an oddball sequence, the successive STD stimuli are readily explained away in nonlemniscal IC neurons [45,46], most likely constituting one of the lowest processing levels in the auditory system of the hierarchical generative model [43,44]. However, the low predictability of DEV stimuli would lead to some irreducible PE. The random 10% chance of getting a DEV stimulus adds uncertainty to the expectation of a STD repetition, and such uncertainty must be accounted for in order to minimize this otherwise irreducible PE during perceptual inference. Higher-order auditory processing levels are endowed with larger timescales of integration and only receive the bottom-up PEs reporting the rare and aleatory appearances of DEV stimuli. Hence, the probabilistic structure of the oddball paradigm and its volatility could be encoded at higher auditory stations, which may, in turn, backward signal the expected precision at lower processing levels by means of neuromodulation. Thus, as hierarchically higher processing levels somewhat foresee and top-down communicate the stochastic 10% possibility of failing in the STD prediction, the appearance of a DEV stimulus in the middle of a train of STD stimuli will not be as surprising, thus attenuating PE signaling.

Following this rationale, dopamine release in the auditory midbrain would modulate the postsynaptic transmission of surprising sensory information by encoding uncertainty in the auditory context. As a result, the otherwise irreducible PE prompted by oddball stimuli can be minimized from a rather early stage of auditory processing. In natural conditions, such midbrain-level scaling of PEs could be performed by SPF dopaminergic neurons projecting to the IC [51–58]. Both glutamatergic and GABAergic neurons in the nonlemniscal IC receive comparable dopaminergic innervation from the SPF [55]. Thereby the SPF could finely modulate the postsynaptic gain of both excitatory and inhibitory activity in the auditory midbrain. This top-down adjustment of expected precision would be done in conjuncture with other neuromodulatory systems; e.g., cholinergic projections synapse with glutamatergic neurons in the nonlemniscal IC [55], yielding modulatory effects, which seem to complement those of dopamine [65] (Fig 6).

Many higher and lower nuclei project to the SPF, including the auditory cortex, auditory thalamus, the superior olivary complex, and the IC itself, thus providing the SPF with rich auditory information [56,82–85] (Fig 6, in pink). Outside the auditory pathway, other centers performing higher-order functions in sensorimotor processing and integration also send projections to the SPF, such as the medial prefrontal cortex (Fig 6, in blue) or the deep layers of the superior colliculus [84]. The dopaminergic activity of the SPF must be interwoven to a great extent with the general functioning of the auditory system, thus enabling a potential role in perceptual inference. The reciprocal connectivity of the SPF with many nuclei at multiple levels of the auditory pathway and beyond could provide the neuroanatomical substrate for an early top-down modulation of expected precision, scaling PEs forwarded from the auditory midbrain through descending dopaminergic projections (Fig 6, in purple).

## Limitations

At first glance, our proposal might seem at odds with some previous works from the predictive processing framework regarding neuromodulation. Whereas cholinergic and NMDA manipulation are often reported to yield precision-weighting effects [86–88], dopaminergic effects on PE are less common in the literature [89], and those are mostly linked to processes of active inference [24,34–37], rather than perceptual inference [90]. Besides, classic neuromodulators are often thought to increase the expected precision of PE signaling [86], contrary to the attenuating net effects of dopamine that we found in the nonlemniscal IC. Notwithstanding, it is important to keep in mind that the current view on the relationship between neuromodulation and expected precision derives from (and mainly refers to) cortical data. Cortical intrinsic circuitry and its neuromodulatory sources are vastly different to those of the nonlemniscal IC, so functional differences could be expected as well.

Cortical predictive processing implementations have proposed specific hypotheses about the neuronal types and mechanisms encoding precision-weighted PEs in a defined canonical microcircuit [29,31,32]. Unfortunately, to the best of our knowledge [47,49], current understanding on the intrinsic circuitry of the nonlemniscal IC does not allow for the direct import of such hypotheses into auditory midbrain processing or for us to be more specific about how this mechanism of PE scaling could be implemented (Fig 6). The observed heterogeneity of dopaminergic effects in our sample, in addition to the fact that the SPF dopaminergic projections synapse with both glutamatergic and GABAergic neurons [55], hints at distinct neuronal types fulfilling differentiated processing roles in the proposed precision-weighting mechanism of the nonlemniscal IC. However, it is not possible to distinguish between neuronal types and assign them putative roles solely based on their functional data [91]. So far, not even the mechanisms underlying SSA in the auditory midbrain are clearly established. A recent study using in vivo whole-cell recordings in the IC found that excitatory and inhibitory inputs of SSA neurons did not differ significantly from other neurons not exhibiting SSA, and that synaptic adaptation could not be used to predict the presence of spiking SSA [92].

Finally, microiontophoresis cannot reproduce the temporal and spatial aspects of natural dopamine release, mimicking the tonic and phasic firing of dopaminergic neurons. Applied dopamine may recruit receptors that are not normally activated by endogenous release. The discrepancy between our observed net effect of PE attenuation and the amplifying effect expected by predictive processing models could be caused by such technical limitation. In any case, this study confirms that dopamine can modulate unsigned PE attending to the probabilistic structure of sensory input. Furthermore, our results make room for distinct roles of neuromodulation along the multiple stages of predictive processing along the auditory hierarchy, a possibility that could be further addressed in future studies.

## Conclusions

Our study demonstrates that dopamine modulates auditory midbrain processing of unexpected sensory input. We propose that dopamine release in the nonlemniscal IC encodes uncertainty by reducing the postsynaptic gain of PE signals, thereby dampening their drive over higher processing stages. The dopaminergic projections from the thalamic SPF to the IC could be the biological substrate of this early precision-weight mechanism. Thus, despite being usually neglected by most corticocentric approaches, our results confirm subcortical structures as a key player in PE minimization and perceptual inference, at least in the auditory system.

## Materials and methods

### Ethics statement

All methodological procedures were approved by the Bioethics Committee for Animal Care of the University of Salamanca (USAL-ID-195) and performed in compliance with the standards of the European Convention ETS 123, the European Union Directive 2010/63/EU, and the Spanish Royal Decree 53/2013 for the use of animals in scientific research.

### Surgical procedures

We conducted experiments on 31 female Long–Evans rats aged 9–17 weeks with body weights between 150 and 250 g. We first induced surgical anesthesia with a mixture of ketamine/xylazine (100 and 20 mg/kg respectively, intramuscular) and then maintained it with urethane (1.9 g/kg, intraperitoneal). To ensure a stable deep anesthetic level, we administered supplementary doses of urethane (approximately 0.5 g/kg, intraperitoneal) when the corneal or pedal withdrawal reflexes were present. We selected urethane over other anesthetic agents because it better preserves normal neural activity, having a modest, balanced effect on inhibitory and excitatory synapses [93–96].

Prior to the surgery, we recorded auditory brainstem responses (ABRs) with subcutaneous needle electrodes to verify the normal hearing of the rat. We acquired the ABRs using a RZ6 Multi I/O Processor (Tucker-Davis Technologies, TDT) and BioSig software (TDT) before beginning each experiment. ABR stimuli consisted of 0.1-millisecond clicks at a rate of 21 clicks/second, delivered monaurally to the right ear in 10-dB steps, from 10 to 90 decibels of sound pressure level (dB SPL), in a closed system through a speaker coupled to a small tube sealed in the ear.

After normal hearing was confirmed, we placed the rat in a stereotaxic frame in which the ear bars were replaced by hollow specula that accommodated the sound delivery system. We performed a craniotomy in the left parietal bone to expose the cerebral cortex overlying the left IC. We removed the dura overlying the left IC and covered the exposed cortex with 2% agar to prevent desiccation.

### Data acquisition procedures

Experiments were performed inside a sound-insulated and electrically shielded chamber. All sound stimuli were generated using a RZ6 Multi I/O Processor (TDT) and custom software programmed with OpenEx suite (TDT, https://www.tdt.com/component/openex-software-suite/) and MATLAB (MathWorks, https://www.mathworks.com/products/matlab.html). In search of evoked auditory neuronal responses from the IC, we presented white noise bursts and sinusoidal pure tones of 75 milliseconds duration with 5-millisecond rise-fall ramps. Once the activity of an auditory unit was clearly identified, we only used pure tones to record the experimental stimulation protocols. All protocols ran at 4 stimuli per second and were delivered monaurally in a closed-field condition to the ear contralateral to the left IC through a

speaker. We calibrated the speaker using a ¼-inch condenser microphone (model 4136, Brüel & Kjær) and a dynamic signal analyzer (Photon+, Brüel & Kjær) to ensure a flat spectrum up to approximately 73 dB SPL between 0.5 and 44 kHz and that the second and third signal harmonics were at least 40 dB lower than the fundamental at the loudest output level.

To record extracellular activity while carrying out microiontophoretic injections, we attached a 5-barrel glass pipette to a hand-manufactured, glass-coated tungsten microelectrode (impedance of 1.4–3.5 MΩ at 1 kHz), with the tip of the electrode protruding 15–25 μm from the pipette tip [97]. We place the electrode over the exposed cortex, forming an angle of 20˚ with the horizontal plane toward the rostral direction. Using a piezoelectric micromanipulator (Sensapex), we advanced the electrode while measuring the penetration depth until we could observe a strong spiking activity synchronized with the train of searching stimuli.

Analog signals were digitized with a RZ6 Multi I/O Processor, a RA16PA Medusa Preamplifier, and a ZC16 headstage (TDT) at 12-kHz sampling rate and amplified 251×. Neurophysiological signals for multiunit activity were band-pass filtered between 0.5 and 4.5 kHz. Stimulus generation and neuronal response processing and visualization were controlled online with custom software created with the OpenEx suite (TDT) and MATLAB. A unilateral threshold for automatic action potential detection was manually set at about 2–3 standard deviations of the background noise. Spike waveforms were displayed on the screen and overlapped on each other in a pile-plot to facilitate isolation of units. Recorded spikes were considered to belong to a single unit when the SNR of the average waveform was larger than 5 (51% of the recorded units).

## Stimulation protocols

For all recorded neurons, we first computed the FRA, which is the map of response magnitude for each frequency/intensity combination (Fig 2). The stimulation protocol to obtain the FRA consisted of a randomized sequence of sinusoidal pure tones ranging between 0.7 and 44 kHz, 75 milliseconds of duration with 5-millisecond rise-fall ramps, presented at a rate of 4 Hz, randomly varying frequency and intensity of the presented tones (3–5 repetitions of all tones).

**Protocol 1.** In a first round of experiments, we used the oddball paradigm (Fig 5A) to study SSA. We presented trains of 400 stimuli containing 2 different frequencies ($f_1$ and $f_2$) in a pseudorandom order at a rate of 4 Hz and a level of 10–40 dB above threshold. Both frequencies were within the excitatory FRA previously determined for the neuron (Fig 2) and evoked similar FRs. One frequency ($f_1$) appeared with high probability within the sequence (STD; $p = 0.9$). The succession of STD tones was randomly interspersed with the second frequency ($f_2$), presented with low probability within the sequence (DEV; $p = 0.1$). After obtaining 1 data set, the relative probabilities of the 2 stimuli were reversed, with $f_2$ becoming the STD and $f_1$ becoming the DEV (Fig 5A). This allows to control for the physical characteristics of the sound in the evoked response, such that the differential response between DEV and STD of a given tone can only be due to their differential probability of appearance. The separation between $f_1$ and $f_2$ was 0.28 (49 units) or 0.5 (45 units; Protocol 2 was also applied to these units) octaves, which is within the range of frequency separations used in other previous studies [45,46,60,64–66,98,99]. The units from those 2 groups were pooled together, because their responses did not differ significantly. DEV and STD responses were averaged from all stimulus presentations from both tested frequencies.

The CSI was calculated as follows:

$$CSI = \frac{DEV_{f1} + DEV_{f2} - STD_{f1} - STD_{f2}}{DEV_{f1} + DEV_{f2} + STD_{f1} + STD_{f2}}$$

where $DEV_{fi}$ and $STD_{fi}$ are FRs in response to a frequency $f_i$ when it was presented in DEV and STD conditions, respectively. The CSI ranges between −1 and +1, being positive if the DEV response was greater than the STD response. The FRs in response to DEV or STD stimuli were calculated using windows of 100 milliseconds starting at the beginning of each stimulus. SFRs were calculated using windows of 75 milliseconds previous to each individual stimulus.

**Protocol 2.** In light of the recent discovery of PE signals recorded in the nonlemniscal IC [45], we decided to adapt our stimulation protocol to that of Parras and colleagues for a second round of experiments, which incorporated the CAS [61]. By arranging a set of 10 tones in a regular succession of ascending or descending frequency, no tone is ever immediately repeated. Consequently, whereas CAS does not induce SSA (as opposed to STD), its pattern remains predictable, so the next tone in the sequence can be expected (as opposed to DEV). Thus, this design contains 3 conditions of auditory transit: (1) no change or predictable repetition (i.e., STD), which is the most susceptible to SSA or repetition suppression (Fig 5A, bottom); (2) predictable change (i.e., CAS; Fig 5B); and (3) unpredictable change (i.e., DEV), which should allegedly elicit the strongest PE signaling when it surprisingly interrupts the uniform train of STDs (Fig 5A, top).

Therefore, after computing the FRA (Fig 2), we selected 10 evenly spaced tones at a fixed sound intensity 10–40 dB above minimal response threshold so that at least 2 tones fell within the FRA limits. Those 10 frequencies were separated from each other by 0.5 octaves, in order to make the results comparable to those of [45]. We used the 10 tones to build the ascending and descending versions of CAS (Fig 5B). We selected 2 tones within that lot to generate the ascending and descending versions of the oddball paradigm (Fig 5A), comparing the resultant DEV with their corresponding CAS versions (Fig 5B). All sequences were 400 tones in length, at the same, constant presentation rate of 4 Hz. Thus, each frequency could be compared with itself in DEV, STD, and CAS conditions (Fig 5A and 5B), obtaining 40 trials per condition. To allow comparison between responses from different neurons, we normalized the spike count evoked by each tone in DEV, STD, and CAS as follows:

$$DEV_N = \frac{DEV}{N}; \ STD_N = \frac{STD}{N}; \ CAS_N = \frac{CAS}{N}$$

where

$$N = \sqrt{DEV^2 + STD^2 + CAS^2}$$

From these normalized responses, we computed the iMM as:

$$iMM = DEV_N - STD_N$$

These indices range between −1 and 1. The iMM is largely equivalent to the classic CSI as an index of SSA, as previously demonstrated by Parras and colleagues (see their S2 Fig in [45]). Nevertheless, please beware the CSI provides 1 index for each pair of tones in the oddball paradigm, whereas the iMM provides 1 index for each tone tested.

## Dopaminergic manipulation procedures

After recording the chosen stimulation protocol in a "control condition," i.e., before any dopaminergic manipulation, we applied either dopamine or the $D_2$-like receptor antagonist eticlopride (Sigma-Aldrich Spain) iontophoretically through multibarreled pipettes attached to the recording electrode. The glass pipette consisted of 5 barrels in an H configuration (World Precision Instruments, catalogue no. 5B120F-4) with the tip broken to a diameter of 30–40 μm [97]. The center barrel was filled with saline for current compensation (165 mM NaCl). The

others were filled with dopamine (500 mM) or eticlopride (25 mM). Each drug was dissolved in distilled water, and the acidity of the solution was adjusted with HCl (pH 3.5 for dopamine; pH 5 for eticlopride). The drugs were retained in the pipette with a current of −20 nA and ejected using currents of 90 nA (Neurophore BH-2 system, Harvard Apparatus). Thus, we released dopamine or eticlopride into the microdomain of the recorded neuron at concentrations that have been previously demonstrated effective in in vivo studies [52]. About 5 minutes after the drug injection, we repeated the FRA and the chosen stimulation protocol continuously until the drug was washed away, leaving roughly 2–3 minutes between the end of one recording set and the beginning of the next one. The recording set showing a maximal SSA alteration relative to the control values was considered the "drug condition" of that neuron. We established the "recovery condition" when the CSI returned to levels that did not significantly differ from control values, never before 40 minutes postinjection. We used either dopamine or eticlopride during Protocol 1, whereas only dopamine was tested during Protocol 2.

## Histological verification procedures

At the end of each experiment, we inflicted electrolytic lesions (5 µA, 5 seconds) through the recording electrode. Animals were euthanized with a lethal dose of pentobarbital, after which they were decapitated, and the brains immediately immersed in a mixture of 1% paraformaldehyde and 1% glutaraldehyde in 1 M PBS. After fixation, tissue was cryoprotected in 30% sucrose and sectioned in the coronal plane at 40-µm thickness on a freezing microtome. We stained slices with 0.1% cresyl violet to facilitate identification of cytoarchitectural boundaries (Fig 1). Finally, we assigned the recorded units to one of the main subdivisions of the IC using the standard sections from a rat brain atlas as reference [100].

## Data analysis

The peristimulus histograms representing the time-course of the responses (Figs 3E, 3F, 4E and 4F) were calculated using 1-millisecond bins and then smoothed with a 6-millisecond gaussian kernel ("ksdensity" function in MATLAB) to estimate the spike-density function over time.

All the data analyses were performed with SigmaPlot (Systat Software, https://systatsoftware.com/products/sigmaplot/) and MATLAB software, using the built-in functions, the Statistics and Machine Learning toolbox for MATLAB, and custom scripts and functions developed in our laboratory. Unless stated otherwise, all average values for trials and neurons in the present study are expressed as "median (interquartile range)," because the data did not follow a normal distribution (1-sample Kolmogorov-Smirnov test).

We performed a bootstrap procedure to analyze dopaminergic effects on each individual neuron. SSA indices, both CSI [40] and iMM [45], are calculated from the averages of the single-trial responses to DEV, STD, and CAS. Consequently, only 1 value of such indices can be obtained for each unit and condition. Therefore, to test the drug effects on each unit, we calculated the 95% bootstrap confidence intervals for the SSA index in the control condition. The bootstrap procedure draws random samples (with replacement) from the spike counts evoked on each trial, separately for DEV and STD stimuli, and then applies either the CSI or the iMM formula. This procedure is repeated 10,000 times, thus obtaining a distribution of expected CSI values based on the actual responses from a single recording. We applied this procedure using the bootci MATLAB function, as in previous studies of SSA neuromodulation in the IC [62,64–66], which returned the 95% confidence interval for the CSI in the control condition. We considered drug effects to be significant when SSA index in the drug condition did not overlap with the confidence interval in the control condition.

We used the Wilcoxon signed rank test (signrank function in MATLAB) to check for differences at the population level between the control and drug CSI and FRs.

## Supporting information

**S1 Data. Effect of dopamine on the FRA of IC neurons.** FRA, frequency response area; IC, inferior colliculus.
(XLSX)

**S2 Data. Effect of dopamine on neuronal responses to an oddball paradigm in the IC.** IC, inferior colliculus.
(XLSX)

**S3 Data. Effect of eticlopride on neuronal responses to an oddball paradigm in the IC.** IC, inferior colliculus.
(XLSX)

**S4 Data. Effect of dopamine on neuronal responses to an oddball paradigm and the corresponding cascade control in the IC.** IC, inferior colliculus.
(XLSX)

## Acknowledgments

We thank Drs. Edward L. Bartlett, Nell Cant, Tetsufumi Ito, Adrian Rees, and Richard Rosch for their useful comments on previous versions of the manuscript. We also thank Mr. Antonio Rivas Cornejo and Ms. María Torres Valles for taking care of histological processing.

## Author Contributions

**Conceptualization:** Catalina Valdés-Baizabal, Guillermo V. Carbajal, David Pérez-González, Manuel S. Malmierca.

**Data curation:** Guillermo V. Carbajal.

**Formal analysis:** Catalina Valdés-Baizabal, Guillermo V. Carbajal.

**Funding acquisition:** Manuel S. Malmierca.

**Investigation:** Catalina Valdés-Baizabal.

**Methodology:** Guillermo V. Carbajal, David Pérez-González, Manuel S. Malmierca.

**Project administration:** Manuel S. Malmierca.

**Supervision:** David Pérez-González, Manuel S. Malmierca.

**Validation:** Manuel S. Malmierca.

**Writing – original draft:** Catalina Valdés-Baizabal, Guillermo V. Carbajal, Manuel S. Malmierca.

**Writing – review & editing:** Guillermo V. Carbajal, David Pérez-González, Manuel S. Malmierca.

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
