## [Editor Report · Decision Letter 0]

8 Nov 2019

Dear Manolo, 

Thank you for submitting your manuscript entitled "Dopamine gates prediction error forwarding in the cortices of the inferior colliculus" for consideration as a Research Article by PLOS Biology. Sorry for the delay incurred while we sought external advice.

Your manuscript has now been evaluated by the PLOS Biology editorial staff, as well as by an academic editor with relevant expertise, and I'm writing to let you know that we would like to send your submission out for external peer review.

Please re-submit your manuscript within two working days, i.e. by Nov 11 2019 11:59PM.

Kind regards,

Roli

Senior Editor

PLOS Biology

---

## [Decision Letter · Decision Letter 1]

10 Dec 2019

Dear Manolo,

Thank you very much for submitting your manuscript "Dopamine gates prediction error forwarding in the cortices of the inferior colliculus" for consideration as a Research Article at PLOS Biology. Your manuscript has been evaluated by the PLOS Biology editors, an Academic Editor with relevant expertise, and by three independent reviewers.

IMPORTANT: You'll see that all three reviewers are broadly positive about your study, but that reviewers #1 and #3 request a significant number of additional analyses, and between them the reviewers ask for several textual improvements. In particular, the Academic Editor asked me to emphasise the need to broaden the framing of the paper for those in both the dopamine and predictive coding fields, and also with our wider readership in mind.

In light of the reviews (below), we will not be able to accept the current version of the manuscript, but we would welcome re-submission of a much-revised version that takes into account the reviewers' comments. We cannot make any decision about publication until we have seen the revised manuscript and your response to the reviewers' comments. Your revised manuscript is also likely to be sent for further evaluation by the reviewers.

We expect to receive your revised manuscript within 2 months. 

**IMPORTANT - SUBMITTING YOUR REVISION**

*NOTE: In your point by point response to to the reviewers, please provide the full context of each review. Do not selectively quote paragraphs or sentences to reply to. The entire set of reviewer comments should be present in full and each specific point should be responded to individually, point by point.

*Re-submission Checklist*

*Published Peer Review*

*PLOS Data Policy*

*Blot and Gel Data Policy*

Best wishes,

Roli

Senior Editor

PLOS Biology

REVIEWERS' COMMENTS:

Reviewer #1:

The Authors present the results of single-neuron recordings in the inferior colliculi of anaesthetised mice (N=31), combining auditory oddball paradigms with microiontophoretic application of dopamine or eticlopride. They observe differential effects of dopamine on responses to auditory deviants, standards, and control sounds. These effects are heterogeneous across neurons but at the population level decrease the amplitude of deviant responses. The findings are interpreted within the predictive coding framework as negative precision modulation.

Overall the manuscript is very clearly written and reports novel findings regarding dopaminergic effects on stimulus-specific adaptation. However, the interpretation of the results is less clear, given the heterogeneity of results across neurons and some methodological decisions. 

Major comments:

1. Rationale: While dopaminergic gain modulation is ubiquitous in cortical and subcortical regions, its role in sensory prediction error signalling is not as obvious as implied by this manuscript. All references cited by the Authors in the introduction (line 123) and discussion (line 629) concern dopaminergic effects on reward prediction errors, not sensory prediction errors. The neural implementation of reward and sensory prediction error signalling may be quite different: for instance, reward prediction errors may be signed (positive or negative; Keller & Mrsic-Flogel, 2018), while sensory prediction errors have been suggested to be unsigned (Bastos et al., 2012). In the sensory prediction error literature, cholinergic or NMDA-ergic effects are more often reported (Moran et al., 2013; Rosch et al., 2019; Auksztulewicz et al., 2018) and dopaminergic manipulations much less common (Todd et al., 2013). Within the predictive coding framework, classical neuromodulators are often thought to increase the precision of prediction error signalling (Moran et al., 2013), contrary to the inhibitory net effects described in this manuscript; dissociations between dopaminergic and cholinergic signalling have also been proposed (Marshall et al., 2016). Taken together, the manuscript would benefit from a more comprehensive discussion of how the current results fit into the other studies on neuromodulatory effects on SSA / mismatch responses (including Authors’ own work, e.g. Ayala & Malmierca, 2015) and the broader theoretical framework.

2. Interpretation: Albeit the net effects (averaged across neurons) show reduced responses to deviants in both protocols (lines 414 and 522), the results are very heterogeneous across units. In protocol 1 (classical oddball), a similar proportion of neurons show decreased and increased CSI (39 vs. 30 respectively) following dopamine application; in protocol 2 (oddball + cascade), the numbers are not reported in text but Figure 6 suggests considerable variability across units. Nevertheless, the Authors conclude that dopamine reduces prediction error signalling, possibly due to reduced precision (as understood in the predictive coding framework). However, on grounds of the predictive coding framework, at least in the cortex, specific hypotheses have been formulated regarding (1) the laminar specificity of prediction error signalling (supragranular layers); (2) the class of neurons encoding gain modulation (pyramidal cells). It is unclear whether or how other types of neurons (e.g. inhibitory interneurons) signal sensory prediction errors and how they might be regulated by neuromodulatory gain control. As such, the study has a potential in identifying previously unknown specific effects of dopamine on sensory prediction error signalling in the IC; however, to this end, variability across neurons should be analysed with respect to the types of cells involved (putative pyramidal cells vs. inhibitory interneurons). One possibility is that the neurons showing increased CSI might be predominantly labelled as putative pyramidal cells, while neurons showing decreased CSI might be labelled as putative inhibitory interneurons. Therefore, I would recommend performing a spike shape analysis to classify cells into these two broad categories, and testing for any differences in CSI effects between these classes.

3. Analysis: The Authors report baseline shifts in protocol 1 (Results, lines 427-436) in addition to effects on baseline-corrected measures. The reported baseline shifts are in line with previous proposals that precision modulation should be visible in the pre-stimulus baseline (Hesselmann et al., 2010). However, how might such robust effects of dopamine on spontaneous activity (+61% and -50% in neurons showing decreased vs. increased CSI) affect the analysis of baseline-corrected data? For instance, is the lack of a significant effect on DEV responses (p = 0.114; line 430) due to a ceiling effect given a strong increase in spontaneous firing? More generally, if I understand correctly, the baseline seems to be simply subtracted per condition from the spike-density function of each neuron (lines 298 onwards); as a result, the reported effects on evoked responses may to some extent depend on the baseline effects. Let’s consider the following scenario (based on Fig. 3E):

- The baseline-corrected spike density peak for DEV is 40 spikes/s in the control condition and 30 spikes/s following dopamine (Fig. 3E). At the same time, there is a baseline firing shift of ca. +60% (line 431)

- A neuron with very sparse spontaneous firing in the control condition (e.g. 3 spikes/s) would accordingly peak at 43 spikes/s without baseline correction; after dopamine, the increased baseline would be 4.8 spikes/s (60% more than 3 spikes/s) and the peak response at 34.8 spikes/s. Therefore, in absolute terms, this neuron would reduce its activity following dopamine

- Another neuron with very fast spontaneous firing in the control condition (e.g. 40 spikes/s) would peak at 80 spikes/s without baseline correction; after dopamine, the increased baseline would be 64 spikes/s and the peak response 94 spikes/s. Therefore, in absolute terms, this neuron would actually increase its activity following dopamine. 

The Authors should consider whether and how their effects might be affected by baseline correction methods, and whether other correction methods (e.g. division or log-normalisation) wouldn’t be more desirable. For completeness, could the Authors also report the baseline effects following eticlopride administration as well as dopamine in protocol 2?

Minor comments: 

- Abstract, line 33: I would suggest rephrasing as “… ‘precision’, which is theoretically encoded by the neuromodulatory (e.g., dopaminergic) systems”

- Introduction, lines 92-93: I would suggest rephrasing as “A neurobiologically informed theory of hierarchical perceptual inference, known as predictive coding…”

- Lines 105-107: While a separation of mismatch responses into the prediction error and repetition suppression components based on responses to deviants and standards (vs. controls) is advocated here by the Authors, other accounts (e.g. Garrido et al., 2010; Auksztulewicz & Friston, 2015) have suggested that the difference between deviance detection and adaptation may rather be linked to differences in mechanisms (connectivity vs. gain) or time scales (inference vs. learning)

- Methods, line 205: which frequency range was used in the FRA recordings and subsequent experiments?

- Line 251: Please reiterate that analyses are based on 40 trials per condition in this protocol

- Line 295: Please clarify whether SEMs are across trials, neurons, or both

- Line 299: What was the bin size for PSTH calculation before smoothing? How was the optimal bin size determined?

- Line 307: Were only excitatory responses considered? If so, what were the criteria for excluding inhibitory responses, and how could this influence the results? For instance, could it be the case that some neurons show an overall sensory inhibition relative to baseline in the control condition, but a release from inhibition following dopamine administration?

- Line 331: “a different neuronal response to DEV and CAS could only be due to the fact that DEV violates a prediction of the perceptual model whereas CAS fulfils it” – isn’t this only true under the assumption that only the immediately preceding stimulus influences the neural response? Cascades and oddball sequences are only matched with respect to one preceding stimulus. 

- Lines 387-389: From this section it is not clear what kind of a follow-up analysis is conducted on these neurons, and whether it constitutes a form of double-dipping.

- Lines 390-392: From this section it is not clear that the regression analysis is done across neurons and which variables are treated as predictors

- Results, line 419: Previously in the methods section FRA has only been described as an initial mapping, not as a repeated measurement

- Lines 444-446: Were the effects of dopamine continuously assessed?

- Line 459: Was there enough statistical power, based on 8 or 15 units, to test for statistical significance?

- Line 491: The analysis of first spike latency is not described in the methods and it is unclear why it is reported in this section but not in the other analyses

- Line 518: Please provide numbers of neurons showing increases, decreases, and stable responses

- Discussion, lines 568-570: These effects were reported as not significant in the results section

- Lines 574: The statistical analysis supporting this statement was not reported in the results section

- Lines 604-606 and 610: Fig. 6C suggests that many neurons also have negative mismatch index (iMM); furthermore, dopamine application brings the prediction error index (iPE) to negative values. The Authors should discuss these findings in more detail, as they go beyond the standard understanding of sensory prediction errors within the predictive coding framework.

- Lines 630-636: This paragraph summarises the results in terms of negative precision modulation, focusing on the net effects (but not on heterogeneity of effects across neurons). However, it does not contextualise these findings in terms of predictions of most theoretical accounts of hierarchical perceptual inference or other neuromodulatory effects reported in the IC (see major point 1). I would suggest to expand the discussion on these topics, as it would enhance the impact of the paper as a contribution to revising the predictive coding framework.

- Figures 3C and 4C: please mark statistical significance (or lack thereof) on the plots

- Figure 5C: please clarify whether error bars are across the 5 neurons only

Typos / grammar:

- Line 150: “processed”

- Line 170: “stimuli”?

- Line 237: “- as opposed to DEV.”

- Line 273: “tested during / in”

- Line 273: “in a few cases”?

- Line 331: “could only be due to the fact”

- Line 580: “in the IC cortices, but…”

- Line 645: “extent”

Additional references:

Auksztulewicz R, Friston K. Repetition suppression and its contextual determinants in predictive coding. Cortex. 2016;80:125–140. doi:10.1016/j.cortex.2015.11.024

Auksztulewicz R, Schwiedrzik CM, Thesen T, et al. Not All Predictions Are Equal: "What" and "When" Predictions Modulate Activity in Auditory Cortex through Different Mechanisms. J Neurosci. 2018;38(40):8680–8693. doi:10.1523/JNEUROSCI.0369-18.2018

Ayala YA, Malmierca MS. Cholinergic Modulation of Stimulus-Specific Adaptation in the Inferior Colliculus. J Neurosci. 2015;35(35):12261–12272. doi:10.1523/JNEUROSCI.0909-15.2015

Bastos AM, Usrey WM, Adams RA, Mangun GR, Fries P, Friston KJ. Canonical microcircuits for predictive coding. Neuron. 2012;76(4):695–711. doi:10.1016/j.neuron.2012.10.038

Garrido MI, Kilner JM, Kiebel SJ, Stephan KE, Baldeweg T, Friston KJ. Repetition suppression and plasticity in the human brain. Neuroimage. 2009;48(1):269–279. doi:10.1016/j.neuroimage.2009.06.034

Hesselmann G, Sadaghiani S, Friston KJ, Kleinschmidt A. Predictive coding or evidence accumulation? False inference and neuronal fluctuations. PLoS One. 2010;5(3):e9926. Published 2010 Mar 29. doi:10.1371/journal.pone.0009926

Keller GB, Mrsic-Flogel TD. Predictive Processing: A Canonical Cortical Computation. Neuron. 2018;100(2):424–435. doi:10.1016/j.neuron.2018.10.003

Marshall L, Mathys C, Ruge D, et al. Pharmacological Fingerprints of Contextual Uncertainty. PLoS Biol. 2016;14(11):e1002575. Published 2016 Nov 15. doi:10.1371/journal.pbio.1002575

Moran RJ, Campo P, Symmonds M, Stephan KE, Dolan RJ, Friston KJ. Free energy, precision and learning: the role of cholinergic neuromodulation. J Neurosci. 2013;33(19):8227–8236. doi:10.1523/JNEUROSCI.4255-12.2013

Rosch RE, Auksztulewicz R, Leung PD, Friston KJ, Baldeweg T. Selective Prefrontal Disinhibition in a Roving Auditory Oddball Paradigm Under N-Methyl-D-Aspartate Receptor Blockade. Biol Psychiatry Cogn Neurosci Neuroimaging. 2019;4(2):140–150. doi:10.1016/j.bpsc.2018.07.003

Todd J, Harms L, Schall U, Michie PT. Mismatch negativity: translating the potential. Front Psychiatry. 2013;4:171. Published 2013 Dec 18. doi:10.3389/fpsyt.2013.00171

Reviewer #2:

This manuscript describes the effect of Dopamine release on the response magnitude of unpredictable but not predictable stimulus events. The results are discussed in the framework of the propagation of prediction errors from the midbrain to higher stations and suggests a gating-role of Dopamine activity.

 A stimulus-specific adaption (SSA) paradigm was used, i.e., presenting a tone sequence containing predictable (high probability; standard) tones and, intermixed, some unpredictable (low probability; deviant) tones of a slightly different frequency than the standard. The authors show that the usual significant increase in response magnitude to deviants is reduced in the presence of dopamine in the cortex of the inferior colliculus. By contrast, the responsiveness to the standard tones is only minimally affected by the presence of dopamine. The experimental findings appear solid and the approach is based on detailed previous studies, many by the current authors, with a set of reasonable controls. 

The main issue with the manuscript is the sole interpretation of the findings in a framework of predictive coding theory. While it is not unreasonable to cast the main hypotheses in such a scheme, it is not helpful to omit potential consequences or other potential interpretations since Dopamine release has been postulated to participate in a number of subcortical and cortical functions. Dopamine neurons have been shown to be activated by new stimuli or unpredicted rewards and reporting an error in the prediction of reward during learning. They also play a role in long-term potentiation, a putative cellular mechanism underlying plasticity, and in controlling membrane potential states. The proposed role of dopamine in modulating or even gating propagation of prediction error signals from the IC to higher stations seems worthwhile, yet a number of related aspects remain unclear. For example it is not made clear how error encoding “precision” is related to changes in response magnitude to the deviant (the implicit assumption that firing follows a strike poisson rule is not necessarily valid). Furthermore, the Dopamine is experimentally released minutes before the stimulus sequence, thus it is not related in time to a singular deviant event but is more akin to a local state-change as in a naturally slow release of Dopamine due to unknown causes. These aspects would make a more thorough discussion of the findings worthwhile for the reader.

In the following are a number of questions/points that would be useful to address.

Line 95: “….explains predictable inputs away.” To understand this phrase requires a thorough knowledge of predictive coding theory. Of course the stimulus-based information – in addition to the expectation/predictability information – is still present.

Line 97: “superior … levels”. Should be ‘proximal’ or ‘higher’

Line 116: unclear meaning of sentence: “….to adjust the relative influence of prediction error signals over perception”. Perception is not unequivocally tied to stimulus or error representation.

Line 120: unclear meaning - reformulate: “regulates forward message-passing modulating the postsynaptic gain”

Line 127: perhaps “…modulates response properties and possibly predictive auditory processing aspects”.

Lone 129: “perhaps: “….IC cortices, compatible with the notion of lowering the precision of prediction signals”. But see comment above about magnitude strength and variance.

Line 224: What are potential consequence for using 0.28 octave spacing in Protocol 1 and 0.5 oct in Protocol 2? 

Line 233: rewrite “… a bunch of tones…”; too colloquial

Line 554: the word “gating” usually implies an all-or-nothing mechanisms.

Line 604: State the reason why you conclude this (because CAS responses were not affected….)

Line 615: One could add here that the differential effect of Dopamine on DEV and STD can not be explained by differences in the initial firing rate for DEV and STD because CAS has also a relatively high firing rate and seem unaffected.

Lines 630 to 636: largely speculative and not very crisp. Especially the last line “….deeming their ‘reliability’…..” is very opaque.

Reviewer #3:

This is a nice manuscript that looks at the role of dopamine and how it modulates stimulus specific adaption at the level of the auditory midbrain. The authors show that dopamine signaling modulates SSA in the IC cortices and that this signaling is highly correlated with and consistent with a prediction error signal (top down modulation). Below I provide comments / suggestions, particularly with details that need clarification or could be improved for correctness. 

1) The authors performed histological verification with lesions to verify that they are in cortical regions of IC. It would be highly valuable to know where exactly in the IC cortices these recordings are situated. Are the recordings in dorsal cortex, lateral cortex etc? Can the authors collapse their location information on a standardized atlas of IC? And are there differences in location between sites that exhibit significant changes with dopamine / or antagonist? I realize that a full mapping of these functional properties may be a bit beyond the scope if this initial manuscript, having slightly more detailed anatomical information as it relates to the functional properties would be valuable nonetheless.

2) Line 33 and 44 – abstract – can the authors more concisely define their usage of “precision” or alternately use a more suitable description. Its not obvious how being accurate (precise) is the relevant factor in this study. What seems to be most important is simply the fact that dopaminergic system modulates neural response in a manner the mimics prediction error signaling. This is precisely what the study shows. As far as I can tell, I don’t really see anything in the manuscript regarding precision, since the authors are not using any measures that quantify how accurate the signaling really is.

3) Line 196 – “spike waveforms are identical and clearly separable from other smaller units and the background noise …”

First, the waveforms may very well be very consistent and similar across spike but there is no way that they can be identical!

Have the authors quantified the waveform quality? And did you use a quantitative selection criteria other than visual inspection (not ideal) to designate single units? For example, one can compute the signal-to-noise ratio (or a multitude of other quality metrics) of the spike waveforms and can designate single units as those waveforms that meet a specific criteria (for instance SNR>5).

At minimum, it would be valuable if the authors compute a metric such as SNR for all the units and provide some statistics for the quality of their waveforms.

4) Line 235 and several other locations in the MS – “in consequence” – better to use “Consequently”

5) Line 233 – “bunch of tones”, maybe better to say a “set of 10 tones” or “sequence of 10 tones”. 

6) Line 266 – “microdomaine” – Is there any information the authors can provide or can you estimate the radius of these microdomains? Or even the volume of the drug delivered? It would be valuable to know the spatial extent over which the antagonist affects the IC neural circuitry.

Also, in this regard, can the authors speak towards whether the volume injection size correlates with whether a significant effect is observed or not? Im not sure this is even possible to do, but if it were it would be interesting to know.

7) Line 315 – CSI metric – If one were to compute a CSI for each sound (CSI_1, CSI_2) how does this traditional CSI compare? Is it generally correlated with both? Or does generally one or the other sound dominate? 

8) Line 382 – bootstrap procedure – a bit more detail on exactly how this is done would be useful. Are you bootstrapping across trials? By requiring that the 95% confidence intervals don’t overlap, is this equivalent to a significance criteria of p<0.05?

---

## [Decision Letter · Decision Letter 2]

24 Apr 2020

Dear Manolo,

Thank you for submitting your revised Research Article entitled "Dopamine modulates prediction error forwarding in the nonlemniscal inferior colliculus" for publication in PLOS Biology. I have now obtained advice from two of the original reviewers and have discussed their comments with the Academic Editor. 

Based on the reviews, we will probably accept this manuscript for publication, assuming that you will modify the manuscript to address the remaining points raised by the reviewers and the additional comments from the Academic Editor. Please also make sure to address the data and other policy-related requests noted at the end of this email.

IMPORTANT:

a) Please address the remaining requests from the reviewers.

b) Please address the request from the Academic Editor to make the implications of your findings more accessible to the broad readership of PLOS Biology. This is crucial.

c) ...and please feed this clarity and accessibility through to the title. At the moment, non-neuroscientists will not realise that (while your paper may have wider implications) the focus on the IC means that your study is about the auditory modality. Also they may not appreciate that "prediction error" is more interesting than it sounds. Any improvement to the clarity/accessibility of the title would be appreciated.

d) Please attend to my Data Policy requests below. Your current Data Availability Statement says "data details are available from the authors upon reasonable request" - this is not compatible with PLOS policy.

We expect to receive your revised manuscript within two weeks. Your revisions should address the specific points made by each reviewer. In addition to the remaining revisions and before we will be able to formally accept your manuscript and consider it "in press", we also need to ensure that your article conforms to our guidelines. A member of our team will be in touch shortly with a set of requests. As we can't proceed until these requirements are met, your swift response will help prevent delays to publication.

*Copyediting*

*Published Peer Review History*

*Early Version*

*Submitting Your Revision*

Best wishes,

Roli

Senior Editor

PLOS Biology

ETHICS STATEMENT:

-- Please include the full name of the IACUC/ethics committee that reviewed and approved the animal care and use protocol/permit/project license. Please also include an approval number.

-- Please include the specific national or international regulations/guidelines to which your animal care and use protocol adhered. Please note that institutional or accreditation organization guidelines (such as AAALAC) do not meet this requirement.

-- Please include information about the form of consent (written/oral) given for research involving human participants. All research involving human participants must have been approved by the authors' Institutional Review Board (IRB) or an equivalent committee, and all clinical investigation must have been conducted according to the principles expressed in the Declaration of Helsinki.

DATA POLICY:

Regardless of the method selected, please ensure that you provide the individual numerical values that underlie the summary data displayed in the following figure panels as they are essential for readers to assess your analysis and to reproduce it: Figs 1, 2, 3, 4. NOTE: the numerical data provided should include all replicates AND the way in which the plotted mean and errors were derived (it should not present only the mean/average values).

REVIEWERS' COMMENTS:

Reviewer #1:

Following revisions, the manuscript has improved considerably. I only have a few minor comments:

- In the discussion (lines 530-538), the Authors interpret their finding in terms of dopaminergic signalling of "imprecision" via net negative gain. This seems to be the main interpretation of their results, with major implications for predictive coding theories. However, it is unclear how such net negative gain could be signalled. Could the Authors speculate on a plausible neuroanatomical circuit mediating net negative gain, for example in terms of inhibitory and disinhibitory circuits in nonlemniscal thalamus (Chen et al., JNeurosci 2018, doi: 10.1523/JNEUROSCI.2173-17.2018)?

- In the introduction, the term "negative gain" (line 139) should be clarified

- I would suggest foreshadowing the discussion of sensory vs. reward prediction errors in the introduction so that the reader has a more informed impression of the literature

Reviewer #3:

The authors have adequately addressed all of my comments. Insofar as my first point regarding histological verification, I think it's worthwhile including the histological/lesion picture as supplement material and I encourage the authors to do so. Otherwise, I am fully satisfied with all other aspects of the study. Great Job!

COMMENTS FROM THE ACADEMIC EDITOR:

I’m not quite sure they have done the necessary work yet to broaden the appeal for the broader target readership of the journal. I think they owe a paragraph at the beginning, and probably more text later on, that provides context where the mechanisms that they are identifying are of broader significance and interest.

---

## [Editor Report · Decision Letter 3]

3 Jun 2020

Dear Dr Malmierca,

On behalf of my colleagues and the Academic Editor, David Poeppel, I am pleased to inform you that we will be delighted to publish your Research Article in PLOS Biology. 

Early Version

PRESS 

Kind regards,

Alice Musson

Publishing Editor, 

PLOS Biology

on behalf of

Roland Roberts,

Senior Editor

PLOS Biology